# Detection of Aflatoxin B_1_ in Single Peanut Kernels by Combining Hyperspectral and Microscopic Imaging Technologies

**DOI:** 10.3390/s22134864

**Published:** 2022-06-27

**Authors:** Haicheng Zhang, Beibei Jia, Yao Lu, Seung-Chul Yoon, Xinzhi Ni, Hong Zhuang, Xiaohuan Guo, Wenxin Le, Wei Wang

**Affiliations:** 1Beijing Key Laboratory of Optimization Design for Modern Agricultural Equipment, College of Engineering, China Agricultural University, Beijing 100083, China; s20203071198@cau.edu.cn (H.Z.); lyao@cau.edu.cn (Y.L.); gxhnuli@cau.edu.cn (X.G.); 2019307160201@cau.edu.cn (W.L.); 2Key Laboratory of Food Quality and Safety for State Market Regulation, Institute of Food Safety, Chinese Academy of Inspection and Quarantine, Beijing 100176, China; jiabb@caiq.org.cn; 3Quality & Safety Assessment Research Unit, U.S. National Poultry Research Center, USDA-ARS, 950 College Station Rd., Athens, GA 30605, USA; seungchul.yoon@ars.usda.gov (S.-C.Y.); hong.zhuang@usda.gov (H.Z.); 4Crop Genetics and Breeding Research Unit, USDA-ARS, 2747 Davis Road, Tifton, GA 31793, USA; xinzhi.ni@ars.usda.gov

**Keywords:** aflatoxin B_1_ detection, hyperspectral imaging, peanut, micro methods, interaction mechanism

## Abstract

To study the dynamic changes of nutrient consumption and aflatoxin B_1_ (AFB_1_) accumulation in peanut kernels with fungal colonization, macro hyperspectral imaging technology combined with microscopic imaging was investigated. First, regression models to predict AFB_1_ contents from hyperspectral data ranging from 1000 to 2500 nm were developed and the results were compared before and after data normalization with Box-Cox transformation. The results indicated that the second-order derivative with a support vector regression (SVR) model using competitive adaptive reweighted sampling (CARS) achieved the best performance, with R_C_^2^ = 0.95 and R_V_^2^ = 0.93. Second, time-lapse microscopic images and spectroscopic data were captured and analyzed with scanning electron microscopy (SEM), transmission electron microscopy (TEM), and synchrotron radiation-Fourier transform infrared (SR-FTIR) microspectroscopy. The time-lapse data revealed the temporal patterns of nutrient loss and aflatoxin accumulation in peanut kernels. The combination of macro and micro imaging technologies proved to be an effective way to detect the interaction mechanism of toxigenic fungus infecting peanuts and to predict the accumulation of AFB_1_ quantitatively.

## 1. Introduction

Aflatoxin, a toxic secondary metabolite produced mainly by *Aspergillus flavus* (*A. flavus*) [1], has been classified as a Group 1 carcinogen by the International Agency for Research on Cancer (IARC). There are more than 20 types of aflatoxin derivatives, of which aflatoxin B_1_ (AFB_1_) has strong carcinogenicity and teratogenicity [2]. Since peanuts are rich in nutrients such as lipids and proteins, they are susceptible to infection by *Aspergillus flavus*. Taking safety into consideration, many countries have promulgated limits on the content of aflatoxin in foods and animal feeds. In China, the AFB_1_ content may not exceed 20 ppb (part per billion) in food and feed. In addition, the limit of AFB_1_ content is 8 ppb for peanuts as food ingredients in EU. In the US, the USDA stipulates that the AFB_1_ content in food cannot be higher than 10 ppb.

Traditional methods for detecting the AFB_1_ content such as high-performance liquid chromatography (HPLC), thin layer chromatography (TLC), and enzyme linked immunosorbent assay (ELISA) have advantages of low limit of detection (LOD) and good specificity. However, these methods are time-consuming and labor-intensive, as well as being complicated and destructive to samples. As a result, these methods are not suitable to detect AFB_1_ in peanuts in batches at post-harvest in either storage or food processing facilities. Therefore, the development of an online, non-destructive technology for the rapid detection of AFB_1_ contaminated kernels has been sought after by researchers.

In recent years, due to its critical importance, new technologies have been applied to the post-harvest food quality control, and spectroscopic and spectral imaging technologies have been used to detect fungal infections and mycotoxin production. Tao et al. [3] utilized visible and near-infrared (Vis/NIR) spectroscopy to detect the contamination of AFB_1_ in single peanut kernels; they achieved detection accuracies of 90% and 94% for 20 ppb and 100 ppb thresholds, respectively. Their study showed that spectroscopy could detect AFB_1_ contamination in peanut kernels without requiring complex processing steps. However, the study was limited by providing only the mean spectral information of the samples. Hyperspectral imaging (HSI) that produces both spectral and spatial information has been used to detect fungi and mycotoxins. Wang et al. [4] used HSI combined with principal component analysis (PCA) and stepwise factor discriminant analysis to detect maize kernels contaminated with AFB_1_ at concentrations of 10, 20, 100, and 500 ppb. The detection accuracy of the developed HSI model tested on an independent validation set was 88%. Their result indicated that it would be feasible to discriminate AFB_1_ even at a concentration of 10 ppb.

In order to reveal the interaction relationship between *A. flavus* and peanut kernel based on HSI, three micro-scale techniques, that is, synchrotron radiation-Fourier transform infrared spectroscopy (SR-FTIR), scanning electron microscope (SEM), and transmission electron microscopy (TEM), are adopted to facilitate the determination of dynamic interactions between fungi and kernels, such as nutrient loss and AFB_1_ accumulation in peanut kernels.

HSI is mainly used to obtain molecular vibration and co-frequency and frequency multiplication information of hydrogen-containing functional groups, as well as to determine the morphological structure and composition changes of samples. Meanwhile, SR-FTIR yields fundamental frequency information regarding molecular vibrations; thus, it was selected to obtain more detailed spectral information. Additionally, SR-FTIR, which combines a synchrotron radiation source and infrared microscopy, can reach a diffraction limited resolution and achieve high signal-to-noise ratios. Hence, it is extremely suitable for small-sized samples in the microscopic field or large microscopic spectroscopy studies with non-uniform regions [5]. In other words, the other reason we adopted SR-FTIR is due to its microscopic detection ability for slices, which can be a good complement to HSI’s observations at the kernel level. The authors of [6] investigated the possible interaction between chitosan and α-Terpineol using FTIR. The authors noted that characteristic peaks appeared in the α-TCsNe spectral data, confirming the successful encapsulation of α-Terpineol into chitosan nano emulsion.

Furthermore, TEM and SEM were used to more clearly observe the surface morphology and to obtain chemical information of the fungi-kernel complex. Specifically, SEM technology can provide information on the micro-topography of the sample surface and the chemical composition of the sample by extracting signals such as scattered electrons [7]. SEM technology was used to observe the morphology of starch granules from high-amylose rice and wild-type rice [8]. Those authors found that high-amylose rice was similar to amylose potato starch and the starch from the *ae* mutant of maize. TEM projects an accelerated and concentrated electron beam onto a very thin sample, using the collisions to change the direction, thereby generating solid-angle scattering and forming images with different degrees of brightness [9]. TEM was used to prove that approximately all hyphae were inhibited in stripe rust-resistant cultivar compared with susceptible cultivars, which indicated that TEM was suitable to observe the intracellular changes of the cultivar [10].

In summary, the main objectives of this study are to: (1) explore the feasibility of developing technology for the quantitative detection of the AFB_1_ in peanuts; (2) investigate the interaction mechanisms such as nutrient loss and intracellular structure change of kernels during AFB_1_ biosynthesis based on microscopic and spectroscopic technologies (i.e., SEM, TEM, and SR-FTIR); and (3) select key wavelengths related to peanut infected *A. flavus*.

## 2. Materials and Methods

### 2.1. Sample Preparation

The “*Huashi*” peanut variety was selected as the study subject. Kernels of similar size without skin damage were surface sterilized with 1% sodium hypochlorite for 1 min. and then with 75% ethanol for 10 min. before further processing. Samples were soaked and then rinsed with sterile water twice. Then, the surface moisture of the kernels was wiped off with a sterilized gauze. When the spore suspension was prepared, 15 mL of Tween 80 solution with a concentration of 0.05% was added to the activated medium and spores were scraped onto the surface of the medium with a picking ring. Peanut samples which had been dried were prepared in the laboratory by soaking in an *A. flavus* spore suspension (10^6^ spores/mL) for 30 s. Simultaneously, the control group was soaked in a sterile Tween 80 solution with a concentration of 0.05% for 30 s.

A total of 150 peanut kernels were cultured. About 15 inoculated kernels were placed in a round petri dish with a diameter of 90 mm. There were ten duplicate samples in each group. In order to ensure that the kernels were maintained in an environment with sufficient moisture during the cultivation process, each round petri dish was not covered and placed in a square petri dish with dimensions of 100 mm × 100 mm, and about 25 mL of sterile water was added. To ensure safety, the square dish was covered with a lid. The square petri dishes were placed in an incubator and cultivated for 24–120 h at an interval of 24 h, where the temperature and humidity were set to 30 °C and 60%, respectively. Seventy-two samples with similar growth of *A. flavus* were selected for data collection. Eight peanut kernels were selected per day for 5 days to obtain hyperspectral image data, and the remaining four kernels were used for SEM, TEM, and SR-FTIR data collection.

For SR-FTIR data collection, hyphae on samples were removed with alcohol wipes. The samples were then filled with the embedding agent, and a small amount of liquid nitrogen was used to quickly freeze the samples and stop fungal growth. Then, the samples were sectioned using microtome to a thickness of 10 μm. The cut samples were placed on the BaF_2_ substrate for inspection.

For SEM data collection, samples were cut into pieces with dimensions of approximately 3 mm (length) × 2 mm (width) × 1.5 mm (thickness). After the standard sample preparation procedure, i.e., freezing, fixing, and decolorization, samples were observed under SEM to record the changes in the microstructures of the treated and control samples. For TEM data collection, kernel tissue samples were fixed, sectioned into a thickness of less than 100 nm and stained for TEM. For both SEM and TEM data, kernels were cut longitudinally from the surface to the inside of kernel and the observation direction was longitudinal near the surface in order to inspect the gradual growth of hyphae.

The detection of AFB_1_ in peanuts was modified according to the operation of the AflaStar^TM^ R immunoaffinity column. First, a phosphate buffered saline (PBS) buffer (pH 7.4) and acetonitrile/deionized water (84/16, *v*/*v*) were configured. Then, peanuts with hyphae were ground and put in centrifuge tubes. Next, 5 mL of acetonitrile/deionized water (84/16, *v*/*v*) was added, and the mixture was shaken for 2 h. After shaking, 1 mL of the supernatant was taken and mixed with 9 mL of PBS buffer. The diluted mixture (the dilution for 24–48 h of culture was 5 mL; for 72 h was 1 mL; and for 96–120 h 0.1 mL) was measured and added to a syringe. Then, the immunoaffinity column was rinsed twice with 10 mL of PBS buffer and then again with 10 mL of deionized water. After the eluent was completely emptied, methanol was added to the immunoaffinity column for elution. After the air passed through the column, all the eluent was collected. After the collected eluate was filtered, the filtrate was stored in the sample bottle for AFB_1_ quantitative detection.

### 2.2. Instruments

The instruments used for the current study included a short-wave infrared (SWIR) HSI system, a HPLC system, a transmission electron microscope, a scanning electron microscope, and a synchrotron radiation source.

The SWIR-HSI system consisted of an imaging spectrometer (SWIR Spectral Camera SN462086, SPECIM, Oulu, Finland), a motorized linear stage (TSAxx-B, Zolix, Beijing, China), a front lens with a focal length of 19.9 mm (HSIA-OLESMICRO-10, Dualix, Chengdu, China), a light source (LSB-T150, Zolix, Beijing, China), and a computer acquisition system.

For SWIR hyperspectral image acquisition, samples were placed on a Teflon board. The linear stage moved the samples across the field of view of the camera, capturing a series of line-scan reflectance images across 1000–2500 nm. The number of spectral bands was 270. These individual lines of data were then compiled into a single, coherent hyperspectral image that was ready for further processing.

After the hyperspectral images are collected, pixel-level hyperspectral image corrections were performed. Intensity calibration was done with a white reference image measured by a spectralon reflectance panel (HSIA-CT-400×00, Dualix, Chengdu, China) with 99.9% reflectivity and a dark reference image with the lens cap on. Both reference images for intensity calibration were captured before imaging any peanut samples. After measuring the SWIR images, an intensity calibration was done for each measured SWIR image with the below equation:(1)$$R=\frac{I_{m}-I_{d}}{I_{r}-I_{d}}\times R_{r}$$
where *R* is the relative reflectance after calibration, *I_m_* is measured reflectance, *I_r_* is reflectance of the white reference, *I_d_* is reflectance of the dark reference, and *R_r_* is the nominal reflectivity coefficient of the white reference panel. The specific value of *R_r_* was 0.99. Then, a Savitzky-Golay (S-G) smoothing filter with a window size of 9 was applied to the calibrated spectra at each pixel to suppress the spectral noise.

The HPLC system consisted of HPLC (2690, Waters, Milford, MA, USA), a chromatographic column (TC-C_18_, Santa Clara, Agilent, CA, USA) with temperature of 30 °C, and a photochemical derivatizer (HC09004, Huaan Magnech, Beijing, China). The mobile phase was methanol:water (7:3). The flow rate was 1 mL/min, and the injection volume was 20 μL.

The TEM system (HITACHI H-7500, HITACHI, Tokyo, Japan) was used to observe the samples that were sectioned to less than 10 nm. When observing the sectioned peanut kernel samples, the magnification of the TEM system was 1500×.

The SEM system (HITACHI SU8010, HITACHI, Tokyo, Japan) was used to extract the secondary electrons, scattered electrons, and other signals generated by the interaction of the electron beam with samples to obtain information about the micro-topography of the sample surface and the chemical composition inside the sample.

Infrared spectrum data of the samples ranging from 4000–400 cm^−1^ were collected at the Shanghai Synchrotron Radiation Facility (SSRF) using a FTIR spectrometer (Thermo Nicolet 6700, Thermo Fisher Scientific Inc., Waltham, MA, USA) with a Nicolet continuum microscope (Thermo Nicolet IR200/100, Thermo Fisher Scientific Inc., Waltham, MA, USA) [11]. FTIR data were collected using line-scanning mode, which allowed us to collect infrared data line by line. During data acquisition, the spot size of every pixel was 20 μm × 20 μm and the step size of the center of every pixel was 10 μm × 10 μm. The acquisition resolution of the FTIR spectrometer was 4 cm^−1^.

### 2.3. Data Analysis

For each SWIR hyperspectral image, regions of interest (ROI) were created by threshold-based segmentation methods to create mask images using the ENVI 5.3.1 software (Exelis Visual Information Solutions, Boulder, CO, USA). From 1224 nm to 1336 nm, the reflectance of the Teflon board background gradually decreased, while the reflectance of peanut kernels increased first and then decreased. Next, two gray images under bands of 1224 nm and 1336 nm were selected for band math, which means adding, subtracting, multiplying and dividing between different bands. An optimal threshold value obtained by trial and error was applied to segment out the foreground of peanuts to 1 while setting pixels in the background to 0. The mask images obtained from the peanut segmentation process were applied to hyperspectral images.

The obtained spectral data generally contained noise and interference due to the scattering of the peanuts and instrument noise. Therefore, it was important to calibrate and denoise the spectral data before data analysis. In this study, three basic pre-processing methods were applied. Standard normal variate (SNV) was used to eliminate the influences of light scattering and changes in optical path length on spectral data. First-order derivative (1st) and second-order derivative (2nd) were used to eliminate baseline shifts and spectra noise. The high dimensionality of HSI data greatly improves the spectral resolution, providing more accurate information for subsequent analyses. However, the original, unprocessed, high-dimensionality data inevitably contain a lot of redundant information including noise. These data not only increase the number of required calculations, but may also reduce the prediction accuracy, which is known as the dimensional disaster [12]. Compared with the number of samples, the more variables that have a collinear relationship, the more likely it will be that the model is overfitted. The accuracy of the calibration set is high, but it is difficult to achieve a high level of accuracy for independent samples. Therefore, pre-processing and the selection of key wavelengths are advantageous to reduce redundant information and improve the accuracy and stability of the model. As a result, it was necessary to extract key features by reducing the data dimensionality. In this study, three variable selection methods, i.e., competitive adaptive reweighted sampling (CARS), successive projection algorithm (SPA), and weight coefficient selection (WCS), were applied in the selection of spectral features. The CARS combined Monte Carlo sampling and regression coefficients of the partial least squares regression (PLSR) model to select key wavelengths [13]. The SPA selected key wavelengths from redundant data [14]. The WCS selected key wavelengths according to the large PC loading coefficients [15].

The HSI data were divided into a calibration set and a validation set with a ratio of 2:1 using the Kennard-Stone (KS) algorithm. The principle of the KS algorithm is to select two samples with the farthest Euclidean distance into the calibration, and then to find the candidate sample with the largest and smallest distances by calculating the Euclidean distance from each remaining sample to each known sample in the set into until the required number of samples has been reached.

PLSR, one of the most effective chemometric methods, and support vector regression (SVR) with radial basis function (RBF) kernel were used to predict the AFB_1_ concentration. The key idea of using PLSR is to find a linear regression model by projecting all variables (measured AFB_1_ contents and spectral responses) to a new space with reduced dimensionality (i.e., much smaller than the original variable space). The success of SVR is based on the assumption that there will be non-linearity in the hyperspectral data.

Coefficient of determination (R^2^), root mean square error (RMSE), and residual predictive deviation (RPD) were used to assess model performance. R^2^ was used to measure the goodness of fit of a model; the higher R^2^, the better the regression model. RMSE was used to measure the deviation between the true values and the predictions. RPD, the ratio of standard deviation to RMSE, was used to prove the predictive ability of the model.

Linear model analyses usually assume that observations are independent from each other, conforming to the normal distribution [16]. In statistical work, it often occurs that the continuous dependent variables do not conform to the normal distribution. If this is the case, a nonlinear transformation can improve the output [17]. In the current study, the AFB_1_ content of peanut samples was converted by Box-Cox transformation according to the following equation. Box-Cox transformation [18] can be used to normalize data without losing the original sequence information, which can significantly improve the normality and linearity of a data sequence and improve the correlation among the data.
(2)$$Y=\left\{\begin{array}{c} \frac{{\left(y\right)}^{\lambda }-1}{\lambda }{\left(\lambda \neq 0\right)}_{,}\\ \log\left(y\right)\left(\lambda =0\right) \end{array}\right.$$
or
(3)$$Y=\left\{\begin{array}{c} \frac{{\left(y+c\right)}^{\lambda }-1}{\lambda }{\left(\lambda \neq 0\right)}_{,}\\ \log\left(y+c\right)\left(\lambda =0\right) \end{array}\right.$$
where *y* is the AFB_1_ reference value, *Y* is the AFB_1_ value after the transformation, *c* is a constant, and λ is the transformation parameter. Since the Box-Cox transformation requires *y* to be non-negative, the first formula applies to the case of *y >* 0 and the second to *y <* 0. It is noteworthy that the unit of AFB_1_ measurement (*y*) is ppb, while the transformed value (*Y*) is unitless.

## 3. Results and Discussions

### 3.1. Statistical Analyses of AFB_1_ Accumulation

#### 3.1.1. AFB_1_ Distribution in Peanut Samples

The accumulation of mycotoxins in peanut kernels increased proportionally with the growth of *A. flavus*, while the accumulation of AFB_1_ in different kernels varied greatly (Figure 1). *A. flavus* cultured for 24 h did not produce AFB_1_. The AFB_1_ content in peanut kernels was low after 48 h of culturing. The accumulation of AFB_1_ in kernels increased sharply after 48 h of culturing. This result suggests that *A. flavus* could infect peanut samples severely within 48 h and synthesize a large amount of AFB_1_ under suitable artificial inoculation conditions.

#### 3.1.2. Spectral Information Extraction

The reflectance spectral data encompassing the spectral region of 1000–2500 nm for all six sampling times (control, 24, 48, 72, 96, and 120 h) are presented in Figure 2. Within each SWIR wavelength range, there were multiple obvious peaks in the spectra of the peanut kernel samples. These bands mainly contained structural and nutritional information about the kernels. Spectral peaks were observed at near 1125, 1470, 1963, and 2371 nm. The peak near 1125 nm corresponded to the second overtone region of CH_2_ and CH_3_. The peak near 1470 nm was the first overtone of OH [19]. The peak near 1963 nm was related to the water content in the peanuts. Peaks indicating the presence of proteins occurred at 2115–2165 nm and oil at 1705–1795 nm [20]. Obviously, in the region of 1000–1400 nm, absorbance gradually increased while it slightly decreased in the range of 1400–2500 nm with AFB_1_ accumulation.

#### 3.1.3. Quantification Analysis Based on Full Wavelengths

The spectral data were divided into a calibration set and a validation set. Thirty-two values were included in the calibration set and 16 in the validation set. The performance of prediction models to quantitatively detect the AFB_1_ content in peanut kernels is shown in Table 1. The results were obtained by the prediction models based on PLSR and SVR using the original reflectance spectra and preprocessed reflectance spectra with SNV, 1st derivative (Deriv.), and 2nd (Deriv.) for calibration and validation. The maximum R^2^ value in the validation set ($R_{v}^{2}$) was 0.57, and the maximum R^2^ value on the calibration set ($R_{c}^{2}$) was 0.87. The RPD value did not exceed 1.44. Spectral preprocessing could not improve the accuracy of the models. These data were used for direct modeling, even though the model accuracy did not meet the requirements, despite the use of a spectral preprocessing method. Thus, the results suggested that we would have to use other methods to achieve better prediction performance.

Figure 3a shows the statistics of the AFB_1_ content distributions in the peanut samples. As shown in the histogram in Figure 3a, the AFB_1_ values were mainly distributed in the range of 0–10^5^ ppb; this mainly corresponds to the control and the kernels with 0 ppb in the first 24 h (about one-third of the total). The density of the AFB_1_ content at different sampling intervals was very different, in that few peanut kernels showed high AFB_1_ concentrations while kernels with low AFB_1_ concentrations were abundant. The skewed distribution of the data points made it difficult to establish a good prediction model.

To solve this problem, a method of transforming the dependent variable *Y* was proposed. By appropriately transforming this, the comprehensive management of the original data could be achieved to meet the assumptions of the regression model as much as possible. To this end, many appropriate methods exist, such as reciprocal transformation, exponential transformation, etc. Additionally, these transformations can be unified by a formula, i.e., Box-Cox transformation, proposed by Box and Cox in 1964. The model is completely based on parameter *λ*, introduced by the estimation of the sample data itself without any prior information, which overcame many drawbacks of the general transformation model to varying degrees, and had flexible parameter forms. Thus, Box-Cox transformation was applied to the AFB_1_ measurement data in this work. Transformation parameter *λ* was 0.0828. Compared with Figure 3a, the max value changed from 7 × 10^5^ to 50, and the AFB_1_ distribution probability after the transformation was more in line with a normal distribution (Figure 3b). The limit of AFB_1_ according to food safety standards in China, 20 ppb, corresponded to the transformed value 3.46 after Box-Cox transformation. Therefore, in subsequent analyses, the transformed value 3.46 was adopted as the new threshold for AFB1 content predictions.

The results of the prediction models established after Box-Cox transformation are shown in Table 2. The results were better compared to the modeling results before the Box-Cox transformation. The $R_{v}^{2}$ values of PLSR and SVR models were between 0.89–0.91 and 0.88–0.94, respectively. Obviously, spectral preprocessing had little effect on the performance of the PLSR model. The best result was obtained by the SVR model with RBF (c = 2.29, g = 0.0089) and 1st derivative spectra. The R^2^ on the calibration and prediction sets were 0.97 and 0.91, respectively. The RPD reached 3.24.

#### 3.1.4. Modeling Results Based on Selected Wavelengths

As mentioned above, CARS, SPA, and WSC were applied to select important spectral variables for the detection of AFB_1_ content, including 10, 7, and 8 wavelengths over the spectral data range of 1000–2500 nm, respectively (Table 3). Instead of using the full spectra, parsimonious models were developed using the key wavelength variables determined by CARS, SPA, and WCS, respectively. The results of the multi-band models using the selected wavelengths are presented in Table 4. The CARS-SVR model achieve the highest $R_{c}^{2}$ and $R_{v}^{2}$ of 0.95 and 0.93, respectively. The CARS-PLSR model (Figure 4) achieved $R_{c}^{2}$ and $R_{v}^{2}$ of 0.94 and 0.92, respectively, The RPD value of CARS-based regression model showed better performance than the SPA- and WCS-based models. The performance of the multi-band models decreased compared with the full-band models. However, the number of variables reduced from 270 to about 10. This meant that the processing speed was greatly improved.

The wavelength at 1173 nm corresponded to the second overtone region of CH_2_ and CH_3_. In addition, a few wavelengths related with the second overtone of C¼O stretching due to intermolecular esterification were located within the range of 1950–2000 nm. The wavelengths near 2123 nm and 2316 nm were related to oil. The wavelengths at 1918 nm and 2361 nm corresponded to the peaks (1963 nm and 2371 nm), as shown in Figure 1.

After inoculating with *A. flavus*, the structure, morphology, and nutrient content of the peanut kernels changed significantly [21], which influenced the spectral signals of the kernels. The key wavelengths were mostly related to the starch, protein, oil, and water contents in the peanut kernels when the multispectral model had been established.

Based on the established optimal model (CARS-1st Deriv.-SVR model), each pixel in the peanut kernel was predicted separately; the prediction results are presented in Figure 5b. The ordinate represents the predicted results according to the Box-Cox transformation; the new threshold is 3.46. Compared with the photo of peanut kernels shown in Figure 5a, the results were in accordance with the real conditions. In the color bar, black represents the low AFB_1_ content in kernels, while red is the opposite. In the visualized prediction map, the control and kernels cultured for 24 h were almost blended with the background, such that mycotoxins were almost non-existent. After being cultured for 48 h, AFB_1_ was detected at a high concentration around the kernels. This observation was consistent with the initial growth of *A. flavus* on peanut pericarp. When the culturing time reached 72 h, it was predicted that more pixels would indicate a higher concentration of AFB_1_. The trend was matched with the fact that AFB_1_ content had increased in the kernels. The kernels cultured for 96 h and 120 h were not distinguishable from each other, as the pixels representing the surface of kernels were almost predicted as contaminated pixels. The spore of *A. flavus* first colonized and germinated on the surface of the kernels. Then, with the growth of hyphae, the fungal community continued to grow and the hyphae broke through the protection of kernel and pericarp, extending inside the kernel. At the same time, the hyphae formed the toxic secondary metabolite, AFB_1_. This visualization map was almost consistent with the growth processing of *A. flavus* and actual accumulation of AFB_1_. The result suggested that the CARS-1^1st^ Deriv.-SVR model would be able to show the differences in the concentration of AFB_1_ or its presence in different areas of the kernels.

### 3.2. Micro-Scale Examination of the Complex of A. Flavus and Kernel

#### 3.2.1. SR-FTIR Results and Analysis

The HSI-based models mainly captured frequency multiplication and co-frequency information. Compared to HSI, SR-FTIR mainly captured the fundamental frequencies of the samples. Therefore, as a micro-scale tool, SR-FTIR could yield AFB_1_ microscale information in kernels.

Both peanut kernels and AFB_1_ have complex physical and chemical structures and can easily absorb spectral signals at similar frequencies. As a result, control kernels and kernels cultured for 120 h as healthy and unhealthy kernels were selected to find the wavenumber associated with AFB_1_. Figure 6 shows the infrared spectra of healthy and unhealthy peanut kernels. It shows that the absorption peaks of the two spectra in the functional group area (4000–1800 cm^−1^) were basically the same, but the absorption peaks in the fingerprint area (1800–800 cm^−1^) were mostly different.

The peak in the wavenumber range of 1700–1680 cm^−1^ was due to ketone carbonyl stretching [22]. The vibration mode observed in the wavenumber range of 1290–1270 cm^−1^ could have been due to the C-O-C anti-symmetrical stretching of aromatic acid ester. The unsaturated δ-lactone ring was absorbed at 1270–1060 cm^−1^ [23]. The region between 900 cm^−1^–600 cm^−1^ generally arose from aromatic ring vibrations [24]. In addition, bands centering at 3300 cm^−1^ were attributed to the N–H stretching of amide A of proteins, while those at 2935 cm^−1^ represented the C-H asymmetric stretching of methylene groups of fatty acids, and that at 1750 cm^−1^ was related to C=O stretching of lipids. At 1647 cm^−1^, which corresponds to the absorption peak of amide I, the absorption peak of the healthy kernel was higher than that of the unhealthy one. The absorption peak that appeared near 1535 cm^−1^ was related to amide II [25]. Nutrient contents, i.e., proteins and lipids, in healthy and unhealthy kernels were significantly different. The subtle changes of these absorption peaks indicated the accumulation of AFB_1_.

Using SR-FTIR technology to visually track and analyze AFB_1_ in peanut kernels infected by *A. flavus*, the peanut epidermis tissue was selected as the imaging area, comprising a region of about 200 μm × 150 μm that included the cotyledon and epidermis. According to the above analysis, the characteristic absorption peaks of AFB_1_ (1702 cm^−1^, 1289 cm^−1^, 1072 cm^−1^ and 907 cm^−1^) and nutrients (3290 cm^−1^, 2935 cm^−1^, 1750 cm^−1^, 1647 cm^−1^ and 1535 cm^−1^) shown in Figure 7a,b were selected as the research objects, by tracking, and observing the intensity changes of the absorption peak corresponding to the above wavenumbers in the entire scanning area.

The red in the Figure 7 indicates that the relative absorption intensities of substances in the corresponding area were high, while the blue means that they were weak. The region between 1800 cm^−1^ and 650 cm^−1^ covered the variations that were unambiguous to fungal information [26], which was consistent with the findings that 1702 cm^−1^, 1289 cm^−1^, 1072 cm^−1^, and 907 cm^−1^ were related to AFB_1_ information. In Figure 7a, AFB_1_ track information at 1702 cm^−1^ (ketone carbonyl), 1289 cm^−1^ (aromatic acid ester), 1070 cm^−1^ (unsaturated δ-lactone ring) and 911 cm^−1^ (aromatic ring) are visualized. The appearance of absorption peaks related to AFB_1_ indicated that *A. flavus* had grown in the peanuts, leading to the formation of AFB_1_.

The absorption peaks corresponding to nutrients (i.e., protein and lipid) verified nutrient loss in peanut kernels infected with *A. flavus*, which shown in Figure 7b. The most prominent spectral change was observed as a decrease in peak height of lipid at 1750 cm^−1^. This was a good indicator of *A. flavus* infestation and the decomposition of the peanut kernels [27]. The peak at 2935 cm^−1^ (C-H stretching) also decreased with fungal growth. Proteins amide A (3300 cm^−1^), amide I (1647 cm^−1^), and amide Ⅱ (1535 cm^−1^) appeared in this research. A study by Bothast et al. [27] pointed out that fungus could hydrolyze the proteins of host crops, using the produced building blocks to form their own protein structures in advanced infections. A sharp decrease after the growth of fungus was observed when the amide I and amide II band components were researched. The same trend was observed with the amide A band [24].

#### 3.2.2. Micro-Scale Inter- and Intra-Cellular Analysis

The SEM images contained information about the micromorphologies of the samples, while the TEM images revealed the components in the cells. The combination of SEM and TEM images verified the microscopic changes in the structure and morphology of peanut kernels. As the microscopic images of peanut kernels in Figure 8a(i),b(i) show, the cells in the control group were tightly arranged and nutrient contents, such as proteins, starch, and lipids, were evenly distributed in the cytoplasm. In contrast, Figure 8a(ii–vi), b(ii–vi) shows the microscopic images of the peanut kernels inoculated with *A. flavus* and cultured for 24–120 h with a time interval of 24 h. It was obvious that the cell arrangements had become less compact starting at 96 h (Figure 8a(v),b(v)), and that the distribution of the cell contents had become more disorderly. As shown in Figure 8a(iii),b(iii), 48 h after *A. flavus* inoculation, obvious intercellular spaces began to appear, and the cell contents appeared to be slightly lysed, indicating that the growth of *A. flavus* hyphae and interaction between kernel and *A. flavus* had disrupted the cell wall and deformed the tissue. Figure 8a(iv),b(iv) shows a microscopic image of a kernel at 72 h after *A. flavus* inoculation. It indicates that the cell wall and the cell contents were severely lysed. In addition, spores appeared in the peanut tissue, as shown by the arrow in Figure 8b(iv,v). As shown in Figure 8b(iv), 120 h after the *A. flavus* inoculation, the kernel cell wall was collapsed and broken and nutrients had been consumed. The peanut tissue structure was severely damaged, as indicated in the circled black area in Figure 8b(iv).

## 4. Conclusions

In this study, the interaction mechanism between peanut kernels and mycotoxins was explored using multiple technologies (i.e., HSI, SEM, TEM and SR-FTIR). SWIR-HSI was used to predict the morphological structure and composition changes of peanut kernels infected with *A. flavus*. In order to overcome the serious non-normality of the data distribution of aflatoxin content in the actual samples, the Box-Cox transformation was adopted. The prediction performance of CARS-1st Deriv.-SVR model for SWIR-HSI was the best, with R^2^ of 0.95 and 0.93 for calibration and validation sets, respectively. The key wavelengths related with nutrients in peanuts, such as 1173 nm and 2361 nm, were selected for multi-band SWIR-HSI. Healthy and contaminated kernels could be detected according to the optimal multispectral model established between spectral data and AFB_1_ contents after Box-Cox transformation. Accordingly, the conversion value corresponding to the limit standard of aflatoxin content in food was used as a new threshold. When the predicted mycotoxin value was lower than this threshold, the sample was considered healthy; otherwise, it could be rejected. The composition changes and nutrient loss predicted with SWIR-HSI were evaluated by SR-FTIR. The SR-FTIR technique revealed that the infrared absorption peaks representing nutrients and mycotoxins in the healthy and unhealthy peanut kernels, respectively, were significantly different. A visualization map at these wavenumbers showed the nutrient loss and mycotoxin accumulation of peanut kernels due to infection with *A. flavus*. When peanut kernels were inoculated with *A. flavus* for 24–120 h, changes in structural morphology and nutrients were evident from the combination of SEM and TEM. An analysis of SEM and TEM data also verified that the selection of key wavelengths related to nutrients in HSI was correct. The findings from this study indicate the great potential of using the SWIR hyperspectral imaging to detect AFB_1_ contamination in peanut kernels infected by the *A. flavus*. HSI provides molecular vibration and co-frequency and frequency multiplication information of hydrogen-containing functional groups, and reflects the morphological and composition changes. Fundamental frequency information about molecular vibrations obtained by SR-FTIR proved that composition changes had occurred, as reflected by the HSI. The number of samples studied in this research was limited due to the labor-intensive process of manually crushing the kernels. Although there were 48 samples, these were cultured in 10 different petri dishes, with 15 in each dish. These samples were representative and met the requirement for sample diversity.

This work proposed a new method to detect aflatoxin in peanut kernels. However, there is still a gap in our current knowledge. For example, there was no large-scale sample test. As suggested by the reviewers, the performance of the model proposed in this study could be further improved by increasing the number of samples or checking different types and varieties of peanut kernels and even other crops, such as maize and other nut crops. Additionally, the modeling method used in this work is limited to traditional machine learning. In view of the above problems, our next step will be to develop equipment to complete the large-scale detection and removal of aflatoxin-contaminated crop kernels.

## Figures and Tables

**Figure 1 sensors-22-04864-f001:**
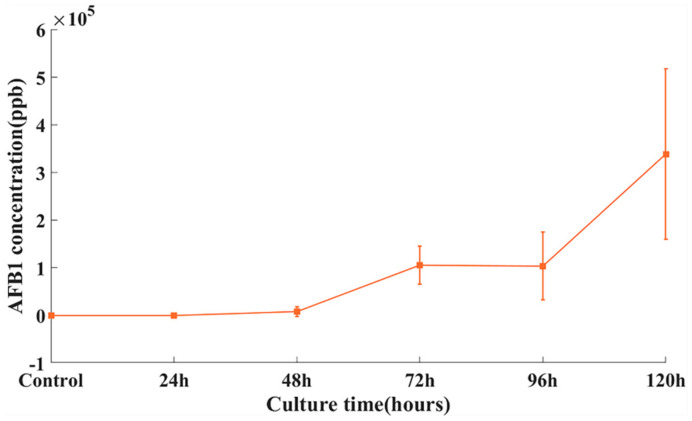
AFB_1_ concentrations of peanut kernels inoculated with *A. flavus* at different time points.

**Figure 2 sensors-22-04864-f002:**
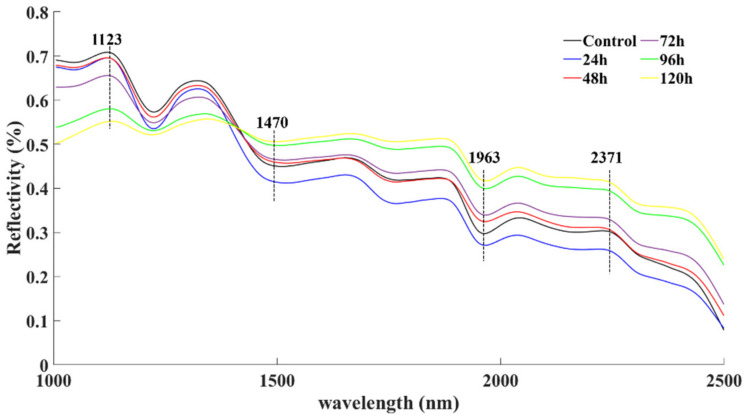
Mean spectral data of peanut kernels inoculated with *A. flavus*.

**Figure 3 sensors-22-04864-f003:**
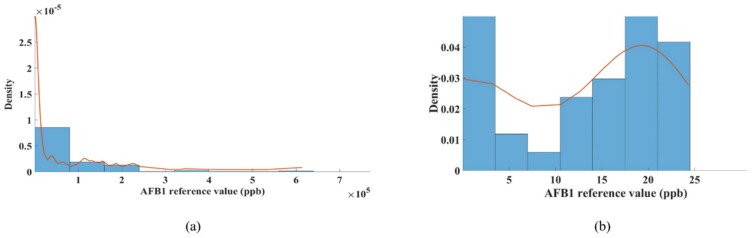
Distribution statistics (**a**) of AFB_1_ in peanut samples and after Box-Cox transformation (**b**).

**Figure 4 sensors-22-04864-f004:**
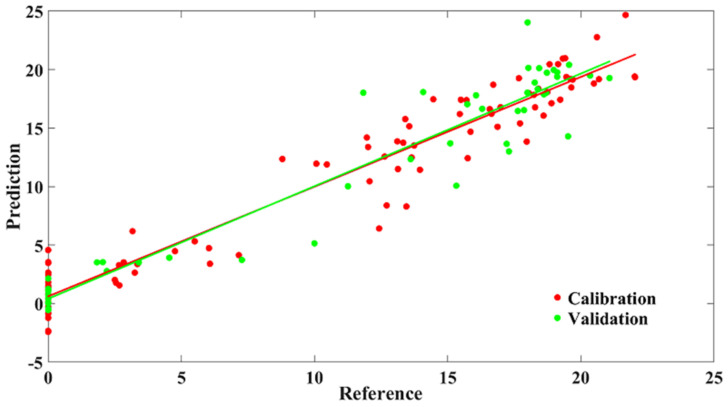
Prediction results of multi-band model.

**Figure 5 sensors-22-04864-f005:**
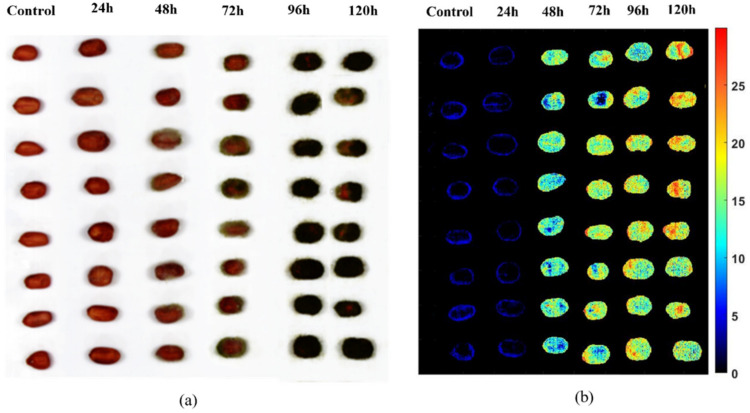
Photograph (**a**) and visualization map of predicted AFB_1_ concentrations (**b**) of peanut kernels.

**Figure 6 sensors-22-04864-f006:**
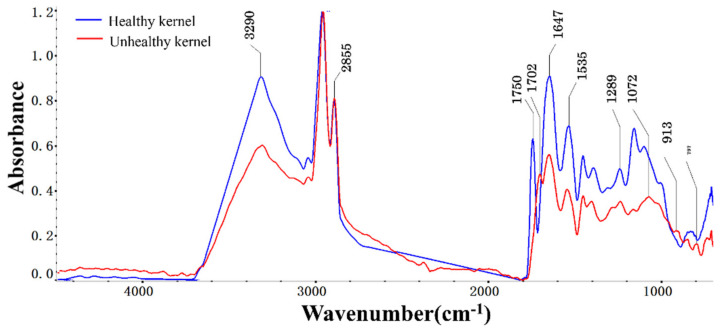
Infrared spectra of healthy and unhealthy peanut kernels.

**Figure 7 sensors-22-04864-f007:**
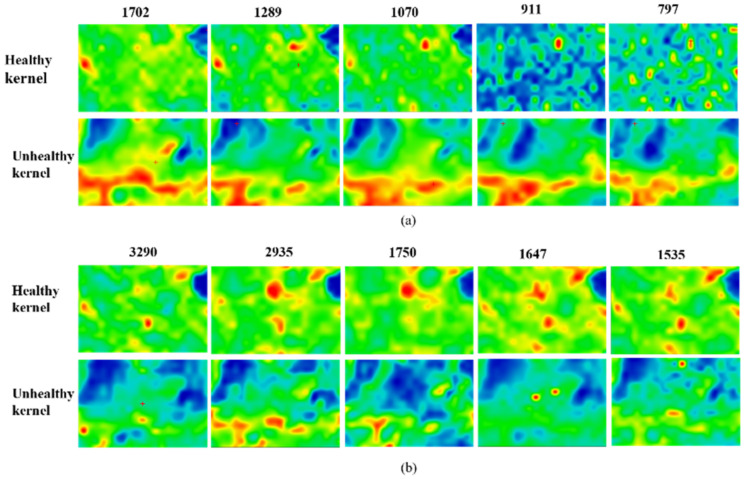
Infrared microscopic images for chemical distributions of trace AFB_1_ accumulation (**a**) and nutrient depletion (**b**) at significant wavenumbers from healthy and unhealthy peanut kernels.

**Figure 8 sensors-22-04864-f008:**
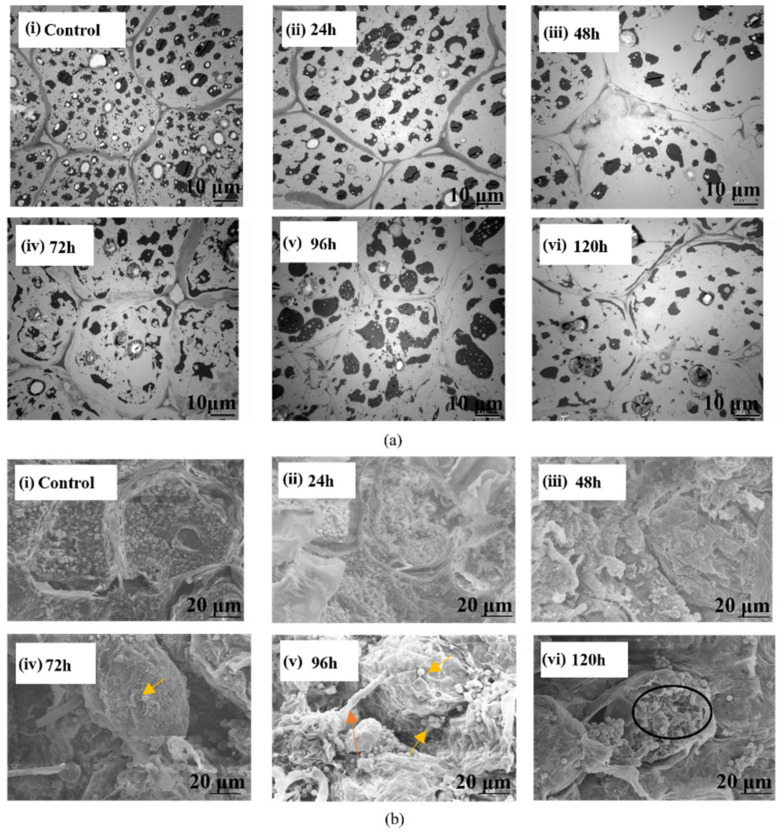
TEM and SEM images of peanut kernel sections inoculated with *A. flavus* (TEM images (**a**) ×1500, SEM images (**b**) ×1000).

**Table 1 sensors-22-04864-t001:** AFB_1_ prediction results of models developed by original AFB_1_ values.

Modeling Method	Pre-Processing	LVs	$R_{c}^{2}$	RMSEc	$R_{v}^{2}$	RMSEv	RPD
PLSR	Reflectance	7	0.65	81424	0.39	126674	1.11
	SNV	9	0.69	95208	0.56	86061	1.30
	1st Deriv.	5	0.59	97577	0.35	99381	1.05
	2nd Deriv.	5	0.68	87665	0.42	121903	1.20
SVR	Reflectance	\	0.54	109734	0.15	147624	0.54
	SNV	\	0.52	119822	0.74	28938	1.84
	1st Deriv.	\	0.53	113386	0.52	80050	0.89
	2nd Deriv.	\	0.86	59764	0.34	111908	1.02

**Table 2 sensors-22-04864-t002:** AFB_1_ prediction results of models with Box-Cox transformation.

Modeling Method	Pre-Processing	LVs	$R_{c}^{2}$	RMSEc	$R_{v}^{2}$	RMSEv	RPD
PLSR	Origin	9	0.96	1.751	0.85	3.582	2.61
	SNV	7	0.93	2.433	0.89	2.650	3.32
	1st Deriv.	8	0.95	2.055	0.87	2.856	2.48
	2nd Deriv.	10	0.97	1.658	0.87	2.957	2.63
SVR	Origin	\	0.96	1.772	0.72	4.916	1.85
	SNV	\	0.95	2.044	0.88	3.205	2.85
	**1st Deriv.**	\	**0.97**	**1.375**	**0.91**	**2.265**	**3.24**
	2nd Deriv.	\	0.99	0.409	0.89	2.798	2.74

Note: Bold indicates that the model is optimal.

**Table 3 sensors-22-04864-t003:** Key wavelengths selected by SPA, CARS and WCS.

Method	Number	Wavelength (nm)
SPA	7	1173, 1580, 1918, 2007, 2338, 2405, 2471
CARS	10	1547, 1630, 1680, 1791, 2067, 2123, 2184, 2316, 2361, 2405
WCS	8	1117, 1307, 1530, 1763, 1946, 2029, 2161, 2422

**Table 4 sensors-22-04864-t004:** AFB_1_ prediction results of models developed by key wavelengths.

Modeling Method	Variable Selection Method	LVs	$R_{c}^{2}$	RMSEc	$R_{v}^{2}$	RMSEv	RPD
PLSR	SPA	5	0.91	2.914	0.90	2.740	2.75
	CARS	6	0.91	2.483	0.90	3.012	3.14
	WCS	6	0.81	3.446	0.79	3.842	1.70
SVR	SPA	\	0.89	3.187	0.87	2.980	2.32
	**CARS**	\	**0.95**	**1.950**	**0.93**	**2.637**	**3.32**
	WCS		0.91	2.337	0.88	2.936	2.53

Note: Bold indicates the optimal model.

## Data Availability

Data is contained within the article.

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
