# Peer review of "Detection of Aflatoxin B1 in Single Peanut Kernels by Combining Hyperspectral and Microscopic Imaging Technologies"

_sensors, 2022, doi:10.3390/s22134864_

Round 1
Reviewer 1 Report
1. The authors used various imaging techniques to explore the mechanism between peanut kernels and mycotoxins.
2. The SWIR hyperspectral imaging used to detect AFB1 contamination in infected peanut kernels seemed to interest the researchers in the same field. However, it would be great if the authors could outline some of the limitations of the current work.
3. The authors stated the possible future work in lines 473 ~ 476. However, it would be great if the possible improvement parameters were briefly discussed such as to mention the "tentative plans to improve the models". This might help the researchers in the same field to get benefit from understanding the overall big picture portrayed by the study.
4. Apart from the above mention elements, the manuscript is well organized and might attract readers in the field.
Thanks.
Author Response
1. The authors used various imaging techniques to explore the mechanism between peanut kernels and mycotoxins.
2. The SWIR hyperspectral imaging used to detect AFB1 contamination in infected peanut kernels seemed to interest the researchers in the same field. However, it would be great if the authors could outline some of the limitations of the current work.
Response: We really appreciate for your positive comments. There are indeed some areas for improvement in our current work. SWIR-HSI represented the molecular vibration and co-frequency and frequency multiplication information of hydrogen-containing functional groups. Although the findings from current study indicated a great potential of using the SWIR hyperspectral imaging to detect AFB1 contamination in peanut kernels infected by the A. flavus, the phenomenon of peak overlap influences the clear display of spectral characteristics for proteins, lipids and carbohydrates. Therefore, the SR-FTIR technology, as a complementary approach in this study, was adopted to prove composition changes from the aspect of fundamental frequency information of molecular vibration. However, in order to make a more detailed analysis, for the SR-FTIR, the microscopic imaging analysis is carried out for the sliced samples. Therefore, unlike hyperspectral imaging, the SR-FTIR as well as SEM are destructive detection techniques that is not suitable for on-line potential applications.
3. The authors stated the possible future work in lines 473 ~ 476. However, it would be great if the possible improvement parameters were briefly discussed such as to mention the "tentative plans to improve the models". This might help the researchers in the same field to get benefit from understanding the overall big picture portrayed by the study.
Response: Thank you for your kind suggestion. It must be admitted that there are still gaps in our current work. For example, the large-scale sample testing does not carry out and the modeling is limited to traditional machine learning. The models described in this study could be improved by increasing the number of samples. In addition, our next step will be to further examine and improve the models by examining different types and varieties of peanut kernels, even other grain and oil seeds, so that to further improve the universality of the model.
4. Apart from the above mention elements, the manuscript is well organized and might attract readers in the field.
Response: Your kind support and warm encouragement are deeply appreciated, it's really an honor for our team. Thank you.
Reviewer 2 Report
In recent years, apart from traditional chemical methods, modern techniques of hyperspectral and microscopic imaging, which are cheaper, faster and, also quite accurate, have been used in research to assess the mycotoxin content in seeds of crops. Therefore, the author´s taking - assess the content of mycotoxins (AFB1) in peanut seeds using several techniques (HSI, SEM, TEM, and SR-FTIR) is correct. The research methods were selected appropriately. The discussion and results were based on the current literature. The inference is correct. The work is interesting, but there is a certain reservation that there are too few samples (48) to assess the amount of AFB1 by SWIR, therefore I propose to supplement them and compare them with the results of other authors' research. In addition, to better fit the model, research on varieties of peanuts and other seeds of crops should be continued, as the authors point out.
The work, after minor changes, is suitable for publication in Sensors.
Author Response
Comments: In recent years, apart from traditional chemical methods, modern techniques of hyperspectral and microscopic imaging, which are cheaper, faster and, also quite accurate, have been used in research to assess the mycotoxin content in seeds of crops. Therefore, the author´s taking assess the content of mycotoxins (AFB1) in peanut seeds using several techniques (HSI, SEM, TEM, and SR-FTIR) is correct. The research methods were selected appropriately. The discussion and results were based on the current literature. The inference is correct. The work is interesting, but there is a certain reservation that there are too few samples (48) to assess the amount of AFB1 by SWIR, therefore I propose to supplement them and compare them with the results of other authors' research. In addition, to better fit the model, research on varieties of peanuts and other seeds of crops should be continued, as the authors point out. The work, after minor changes, is suitable for publication in Sensors.
Response: Thank you for the insightful comment to improve our work. This study is a preliminary exploratory experiment. Considering that this work involves the joint application of a variety of detection tools, especially the complexity of synchrotron radiation sample preparation, we must admit that the final samples used for modeling are indeed limited. Anyway, as you said, the limitation of the samples in this article is indeed a relatively obvious weakness at present. As suggestion in the future work, we will follow the method of this article to examine and improve the models by examining different types and varieties of peanut kernels, even other grain and oil seeds. Thank you very much.
Reviewer 3 Report
This manuscript presents an interesting proposal for identifying the presence of AFB1 accumulation in peanut kernels using hyperspectral imaging.
I think that the topic is very interesting, the text is well written, and the presentation (plots, tablets, etc.) is of high quality. All the methods, results and conclusions are solid, except for the prediction models with PLS and SVMs. In my opinion, the transformation performed to the y values (the prediction parameter measured in ppb) needs much more explanation and discussion, as I find that the results are high (in terms of R2), but I'm not so sure about how this can be translated into real world prediction systems.
I refer the authors to my annotated PDF, as all details are there as comments, handwritten text and others.
and some improvements have been added from previous reviews. Still, the manuscript is not yet suited for its publication as a journal paper, in my opinion. Here are my individual comments:

Author Response
Comments: If this was already published, I would need a better justification of why to use a more expensive equipment (KSI). What is the advantage of using HSI (i.e., using spatial information) than a one-dimensional spectrometer?
Response: AFB1 contamination has a typical heterogeneous distribution. In a batch of contaminated kernel, only a few kernels are observed high concentration AFB1 contamination. Not only that, the AFB1 concentration distribution within the same polluted peanut kernel also show a high degree of local aggregation. Based on visible light/near-infrared (Vis/NIR) spectroscopy, the spatial resolution is limited by the averaging effect of the spectra within the field of view. As a result, it is difficult to achieve effective detection of highly heterogeneously distributed pollutants. HSI technology can collect and process data which contain spatial and spectral information. Although using HSI for the on-line detection of grain is still a difficult issue, it has been successfully applied to the on-line detection of large-scale detection objects, such as fruit, plastic waste recycling and E. coli on the surface of poultry.
Comments: Rr-which specific value in this work?
Response: Thank you for your valuable comment to improve our work. The specific value of Rr is 0.99. And they have been added in Line 177 in the manuscript.
Comments: If the spectral data was so noisy, applying derivative directly would amply this noise, if the signal is not previously filtered (for example, with a S-G filter) was the signal filtered before derivative
Response: Thank you for the kind instructions. I am sorry for not clarifying in Line178-179: First-order derivative (1st) and second-order derivative (2nd) was used to eliminate baseline shift and spectral noise. The S-G smoothing have been adopted in collected HSI 3-D data after hyperspectral image correction. Therefore, the smoothing process does not adopt in the mean spectra selected from ROI again. At the same time, we have checked the spectral of 2nd derivative, no significant noise information appears in the extracted spectral data.
The related results are revised in Line 181-183: Then, the Savitzky-Golay (S-G) smoothing filter with a window size of 9 was applied to the calibrated spectra at each pixel to suppress the objectively existing spectral noise.
Comments: I strongly disagree with this state. I can not think of how the HSI high dimensionality correlates inversely to prediction performance. I actually defend the opposite: the high dimensionality is one benefit that we acquire when using HSI camera, please, elaborate this.
Response: I am sorry for not clarifying in Line: The Hyperspectral images usually have very high dimensionality that cause a degradation in prediction model’s performance.
According to suggestion, the contents have been revised in Line 217-224: The high dimensionality of HSI data does greatly improve the spectral resolution of the data, providing more and more accurate information for subsequent analysis. However, with the increase of spectral dimension, the original unprocessed high- dimensionality data inevitably contains a lot of redundant information irrelevant to the object we want to detect. Thus, these data not only increase the amount of calculation, but also may reduce the prediction accuracy, which is known as the dimensional disaster. Not only that, compared with the number of samples, the more variables that have a collinear relationship, the more likely it will lead to overfitting of the model. That is, the accuracy of the calibration set is high, but the accuracy of independent samples is difficult to satisfy. Therefore, the pre-processing and key wavelengths selection are advantageous to reduce redundant information and improve the accuracy and stability of model.
Comments: Please, elaborate this further. Artificially modify the statistical distribution of the prediction parameter cold indeed improve the model output, how is this addressed when the model is used to do real predictions?
Response: We gratefully appreciate for your valuable questions. In statistical work, especially in the application of linear models, it often occurs that the continuous dependent variables do not conform to the normal distribution. The research in this paper proves that when these data are used for modeling directly, the model accuracy cannot meet the requirement. To solve this problem, the method of transforming the dependent variable Y is proposed. By appropriately transforming the dependent variable Y, the comprehensive management of the original data can be achieved to meet the assumptions of the linear regression model as much as possible. There are many methods for variable transformation of dependent variable Y, such as reciprocal transformation, exponential transformation, etc., and these transformations can be unified by a formula, which is the Box-Cox transformation proposed by Box and Cox in 1964, which is characterized by: The model is completely based on the parameter λ introduced by the estimation of the sample data itself without any prior information, which overcomes many drawbacks of the general transformation model to varying degrees, and has flexible parameter forms.
The limit of AFB1 in food safety standard, 20 ppb was corresponding to 3.46 after Box-Cox transformation. In real prediction, the new threshold (3.46) is adopted for subsequent prediction AFB1 content. In the real scene, when the predicted mycotoxin value is lower than 3.46, the sample is considered healthy, otherwise it is rejected.
Comments: This vale needs to be "de-transformed", does it not? because the model was trained with transformed y's. How should this de-transformation be performed? Please, elaborate and discuss on this. This value should not be named or measured in ppb, as with y, because Y is transformed, and doesn't have the same meaning as the ppb from the original y.
Response: I am sorry for not clarifying here. Firstly, the “de-transformed” process can be performed if people want to know the real AFB1 contents according to equation 1 and 2. Secondly, as mentioned in the previous question, after box-cox transformation, the new threshold (3.46) is applied in subsequent analysis. The healthy and contaminated kernels can be detected directly, therefore it is not necessary to perform the “de-transformed” process. Finally, the unit of AFB1 measurement (y) is ppb while the transformed value (Y) is unitless. The Y value has only statistical significance and no practical significance. However, the classification prediction of samples with different AFB1 values can still be achieved.
Comments: Related to my comments in page 5, this artificial transformation "streches out" the data, the y vales, into a spreaded distribution, and we all know that this increases the performance statistics of the models (especially the R2). Does this reflect a good real prediction capability of the models? Please, address this and discuss.
Response: Thank you for this insightful comment to improve our work. Box-Cox transformation realizes the normalization of data without losing the original sequence information, which can significantly improve the normality and linearity of the data sequence and improve the correlation between the data. Box-Cox transformation has been proved to perform better in AFB1 prediction analysis. Compared with the model established by the AFB1 value without transformation, the Box-Cox transformation improves the normality and linearity of the data, reduces the probability of pseudo-regression, and has higher prediction accuracy and stability. The prediction results meet the detection requirements, it is possible to quickly classify whether the AFB1 contents exceeds the standard.
Round 2
Reviewer 3 Report
While all the new text included in the manuscript is satisfactory, and all the responses were appropriately addressed, all the authors' comments on the statistical data transformation should definitely not only included in the manuscript, but also discussed. If, as the authors state, the use of the trained model in real applications requires to transform it into a classification model (thresholding below a certain 3-point-something value), all this should be included, presented and discussed in the manuscript.
Additionally, I've noticed that a new author has been added. What is the reason for this? The new manuscript has very minor additions (just some scarce text in the introduction), no new models, no new results, no new discussion; so what was the new author's contribution and why wasn't he included in first place?
Author Response
While all the new text included in the manuscript is satisfactory, and all the responses were appropriately addressed, all the authors' comments on the statistical data transformation should definitely not only included in the manuscript, but also discussed. If, as the authors state, the use of the trained model in real applications requires to transform it into a classification model (thresholding below a certain 3-point-something value), all this should be included, presented and discussed in the manuscript.
Response: Thank you for this insightful comment to improve our work. We have added related contents in Line 252-253, 257-259, 264-265, 301-302, 316-326, 329-332, 371-372, 480-482, 486-491, 510-518, 573-574, which shown in the manuscript in blue font.
Additionally, I've noticed that a new author has been added. What is the reason for this? The new manuscript has very minor additions (just some scarce text in the introduction), no new models, no new results, no new discussion; so what was the new author's contribution and why wasn't he included in first place?
Response: Thank you for your carefulness. Because we initially ignored the author's contribution. This author was not added in the original submitted manuscript on 19, May. When submitting the author statement, we reconsidered the contributions of all authors. Considering this author’s contributions in verification and software, with the consent of all authors, we added this author as one of the co-authors. The revised authorship list and author statement were updated on 31 May. However, the original manuscript was sent to you before the revised version was updated in the system. We also explained this issue to the editorial office.